# Virtual Sensor of Gravity Centres for Real-Time Condition Monitoring of an Industrial Stamping Press in the Automotive Industry

**DOI:** 10.3390/s23146569

**Published:** 2023-07-21

**Authors:** Ivan Peinado-Asensi, Nicolás Montés, Eduardo García

**Affiliations:** 1Department of Mathematics, Physics and Technological Sciences, CEU Cardenal Herrera University, C/San Bartolomé 55, 46115 Alfara del Patriarca, Spain; nicolas.montes@uchceu.es; 2Ford Motor Company, Polígono Industrial Ford S/N, 46440 Almussafes, Spain; egarci75@ford.com

**Keywords:** virtual sensor, DBSCAN, predictive maintenance, tonnage monitoring, I3oT

## Abstract

This article proposes the development of a novel tool that allows real-time monitoring of the balance of a press during the stamping process. This is performed by means of a virtual sensor that, by using the tonnage information in real time, allows us to calculate the gravity centre of a virtual load that moves the slide up and down. The present development follows the philosophy shown in our previous work for the development of industrialised predictive systems, that is, the use of the information available in the system to develop IIoT tools. This philosophy is defined as I3oT (industrializable industrial Internet of Things). The tonnage data are part of a set of new criteria, called Criterion-360, used to obtain this information. This criterion stores data from a sensor each time the encoder indicates that the position of the main axis has rotated by one degree. Since the main axis turns in a complete cycle of the press, this criterion allows us to obtain information on the phases of the process and easily shows where the measured data are in the cycle. The new system allows us to detect anomalies due to imbalance or discontinuity in the stamping process by using the DBSCAN algorithm, which allows us to avoid unexpected stops and serious breakdowns. Tests were conducted to verify that our system actually detects minimal imbalances in the stamping process. Subsequently, the system was connected to normal production for one year. At the end of this work, we explain the anomalies detected as well as the conclusions of the article and future works.

## 1. Introduction

In recent years, large companies have been racing towards a full digital transformation, applying technological solutions that allow them to manage the large amount of data they obtain from their businesses. The objective is to obtain solutions that give them a great advantage in decision making and thus increase competitiveness over other companies in the same sector. A full digital transformation has been carried out [1] in many areas, but some of these areas still need to be developed. Fields such as industry or medicine are far from total digitisation due to the complexity of many of their processes. For this reason, a lot of resources are being provided to obtain and develop new technologies in these fields. In industry, progress is being made in what has been called intelligent manufacturing, where applications are being created through the collection of data in order to improve the different phases of the process, from the design to the delivery of the final product. In the literature, one can find a lot of methodologies in different areas of manufacturing [2], such as in the reduction in electricity consumption, where by applying the IIoT technology (industrial Internet of Things) we can obtain, for instance, information on the behaviour of electrical systems of equipment [3] in order to improve the efficiency of both processes and buildings [4]. This can also be achieved by using machine learning techniques [5].

Focusing our attention on maintenance strategies, in recent years, a large number of proposals have been presented regarding the use of sensors and artificial intelligence for defining the state of the machine, also called smart maintenance [6]. As a result, we have found very interesting methodologies such as the case study presented by R. H. Hadi et al., see [7], where the use of automated machine learning techniques (AutoML) has yielded great results when using a large number of functions.

### 1.1. Maintenance Tasks in the Automotive Sector

The automotive world is one of the most demanding, and one of its strengths is a high availability of machines. Many assets are used to achieve an organised production without delays and with the supply of material from controlled suppliers so that there is no error. This is why a breakdown in the manufacturing chain poses great problems, since every minute that no cars are manufactured the company experiences high economic losses. In this way, brands seek to have increasingly modern plants [8] with better connectivity and systems that allow an in-depth knowledge of the state of the process. This is what is intended with the technologies offered by Industry 4.0. Techniques such as the industrial Internet of Things (IIoT), big data analysis and cybersecurity, among others, allow us to develop powerful tools capable of offering solutions demanded by the industry.

### 1.2. Breakdowns in Presses and Major Breakdowns

The stamping process is one of the most complex processes throughout the car manufacturing process, where there are many opportunities for improvement with the use of technologies proposed in Industry 4.0. Industrial presses are a piece of mechanical equipment of great design complexity and large dimensions, where we find a lot of unknown information during the stamping cycle. Production stops due to breakdown can have different causes, since a press is composed of a large number of mechanical elements and a failure of these may cause the equipment to stop. The most serious faults are breaks in the kinematic eccentric transmission system and its components, such as bolts, connecting rods, bearings, gears, etc., due to their high economic impact for a company. That is why it is important to perform proper preventive maintenance. The problem is that even if appropriate maintenance is performed, the breakdowns continue to occur. Even worse, these faults can trigger a major breakdown, with the appearance of cracks in important components, such as connecting rods and eccentric dynamic driven gears, or in the structure of the press, which may incur an exorbitant cost. When these faults appear, it is chaos for the company due to the restructuring that this entails in the plant and the high cost and repair time that these types of breakdowns incur, since the factory does not usually have these types of components in stock and they need to manufacture them.

In the following Figure 1, you can see a triangular connecting rod that has broken transversely following the appearance of a crack. After analysing the transmission system, it was verified that multiple cracks had appeared along the surface of the connecting rod and that this was the consequence of an imbalance or minor fault that ended up causing a crack.

Recent research seeks to obtain new knowledge about what happens with the equipment [9], such as the wear of the tool used [10], and the material in order to avoid two of the great problems that arise throughout manufacturing. These two problems are the breakdown of the different mechanical elements of the press and defects in the final product. The effort made by the press is one of the most-used data sources for the detection of faults in the presses, measured with strain gauge sensors [11,12]. Tonnage data have been treated via different strategies in order to improve the stamping process, such as wavelets [13]. Force sensors are also used to obtain new process characteristics [14,15] by monitoring the data in real time. In [16], a fault prevention approach is proposed for progressive die stamping by using force sensors globally and locally to indicate unhealthy conditions. A similar approach is proposed in [17], where a classification of machine learning algorithms is performed to find out which one of them will make a better prediction.

### 1.3. Previous Works, Mini-Terms

The new methods for maintenance strategies in Industry 4.0 may help decrease the time and cost associated with predictive maintenance. However, there are still challenges to overcome in predictive maintenance [18]. The installation of sensors, their cabling and data extraction through the OT network to the IT network and the increasing number of machines or components to be monitored are some of the key problems. The proposed solutions usually do not end up being implemented in industry in a significant way due to the high cost of their implementation.

In our previous work, we proposed a new methodology based on technologies of the fourth industrial revolution called Mini-Terms [19]. A Mini-Term can be defined as the time it takes for an element of the equipment to perform its work. Thus, by simply programming timers in the PLCs of the manufacturing lines, we can develop a predictive system capable of anticipating failure. The great advantage of Mini-Terms is that they do not not require the installation of any sensors since they use those already installed for the normal operation of the line [20].

The aim of the method is to predict and detect when a sub-component of the manufacturing line is going to break, such as welding guns [21], cylinders, electric motors or bearings, among others, following the philosophy of obtaining all possible information available in the PLC for free without the need for installation of new equipment. At present, there is a standard in the Ford Motor Company on how Mini-Terms should be implemented in PLCs. Currently, more than 34,000 components are being monitored in real time throughout Ford’s factories in Almussafes (Valencia), Cologne and Craiova.

### 1.4. Goal of Our Research

The objective of our research is to create an IIoT+Big Data tool for the stamping process using sensors installed in the press for automatic operation, following the philosophy of our previous works [19,20], and use this information to generate tools in order to improve the process from all points of view: predictive maintenance, energy saving, part quality, etc. For the development of these applications, the first alternative is to directly use the information of the sensors available in the machine but, if this is not possible, the alternative is to try to build a so-called virtual sensor [22], by combining data from different sources, such as physical sensors, other virtual sensors or a combination of both.

In this article, we present a tool that allows us to monitor the balance of the press during the stamping process in real time. The creation of this tool is based on experience acquired by the employees of the factory. It is known that when the mechanical components begin to deteriorate, an imbalance is generated in the mobile part of the press. If the wear of the components is in an advanced stage, this can be identified by the human eye. In the present tool, the imbalance of the press is measured by a virtual sensor which, using the tonnage information in real time, allows a calculation of the gravity centre of a virtual load that moves the mould up and down. The aim is that the gravity centre of the virtual load works as central as possible and without bring over-stressed, thus ensuring that the wear of both the die and the mechanical elements of the press does not increase. The tool developed will be used as a watchdog, notifying us when we are working under unfavourable conditions so that we can make the relevant adjustment to the process and ensure the manufacturing process is being carried out without damage to the press or the tool used.

The article is distributed as follows. In Section 2, a brief description of the stamping process is given. In Section 3, we detail the strategy used to obtain information about the process and the form of the data. In Section 4, we explain the theoretical concept of the virtual sensor method and how to monitor the state of the press taking into account the physical sensors and their advantages. In Section 5 and Section 6, we show how the proposed algorithm has been defined for detecting abnormal working conditions and detail the obtained results. Finally, in Section 7, we show the conclusions and future work.

## 2. Stamping Process

Two types of presses are used in sheet metal forming, mechanical (single action (SA) or double action (DA)) and hydraulic, both can be used in all operations carried out in stamping (blanking, deep drawing and trimming). The major difference between SA and DA presses is the eccentric drive transmission system and the blank holder force system. In SA presses, as seen in Figure 2, there is one slide for the tool and at the bottom of the press there is a cushion system as a blank holder. This is in contrast to DA presses that have a more complex eccentric drive system with two slide displacements, one for the blank holder and other for the punch, but without a cushion at the press base. Double effect presses have been used for years in deep drawing operations, but recently they have been replaced by hydraulic and SA presses with cushions because they are more efficient in deep drawing operations [23].

A cycle of the stamping process can be separated in three phases. First is the downward motion of the slide when the electrical motor activates the kinetic movement of the press through the flywheel with a controlled speed. Second is the stamping phase, where we carry out deep drawing, trimming and bending or flanging and the material is deformed. The last phase is the upward motion where the counterbalance pressure helps the press to rise the die until the end of the cycle, waiting for the idle process to start the cycle again.

In a stamping press, we can find different components that are synchronised. The electric motor activates the movement of the press through the eccentric transmission system. The slide, where the upper die is placed, is moved up and down with a variable press speed due to the design of the eccentric drive system. The hydraulic/pneumatic clutch and brake system start and stop the sliding movement between operations. The pressure of the counterweight helps the electric motor lift the slide. The hydraulic cushion controls the amount of material introduced into the die during stamping. Each of them is composed of several subcomponents such as gears, shafts, connecting rods, pumps, etc. Due to the number of existing items that may fail, the maintenance of a press becomes an arduous task which consumes many resources of the company. Despite the effort made, unexpected breakdowns continue to occur.

## 3. Process Data Acquisition in Press Shops, Smart IIoT

The concept introduced in [19,20] regarding the IIoT generated a new paradigm from the point of view of application development and research to generate industrialisable IIoT tools. The use of sensors, PLC and, in general, the infrastructure already installed and used for production means that any tool developed under this paradigm is directly implementable in the industry and scalable at a very low cost. In [24], this philosophy is defined as I3oT, (industrialisable industrial Internet of Things) since, through the use of what is already installed, tools can be generated with a very low impact and this can generate great benefits.

### 3.1. Criterion 360 (C-360)

In [24], the so-called Criterion 360 for the stamping process was presented. This new criterion creates specific measures for the press since it allows us to obtain 360 data of a variable chosen from the process of a work cycle. The name “criterion” was given due to the fact that today it is only a proposal that aspires to be a standard in the Ford Motor company as Mini-Terms are also present.

The purpose of C-360 is to store data from a sensor each time the encoder indicates that the position of the main axis has rotated by one degree. Since the main axis makes a turn in a complete cycle of the press, this criterion allows us to obtain information from the three phases of the process and easily shows where the measured data are in the cycle measurement data. As a general rule, the degree 180 is the bottom dead centre (BDC) and the degree 320 is the top dead centre (TDC). As seen in the example of the measured data of a tonnage sensor in Figure 3, we can see the moment when the press goes up and down within the cycle. The sensors for which data are being received in the c-360 are shown in Table 1.

The time it takes for the press to run a cycle is about 3–4 s. If the speed of movement of the press is constant, a data point would be stored approximately every 10 ms. However, as we can see in Figure 4, the distance to the BDC of the slide is not linear with respect to the degree of the press position. We can see clearly that in deep drawing, the slide moves more slowly, while in the downward movement, the slide moves with a controlled speed and the highest speed can be observed in the upward movement. Using the C360 criterion implies that for scanning the data in the time domain, the storage frequency of the data varies throughout the cycle. However, when scanning the data in the position domain of the press, the data storage frequency is the same during the cycle.

In the deep drawing stage, the movement is between 150 and 200 mm depending on the die. Here, we can read between 80 and 100 data values, i.e., one every 2 mm. As a result of the fact that the speed of the press is slower here, we have more time to store the data in the PLC die so that we can ensure a higher accuracy when reading the sensor data. In addition, tests were performed to verify the capacity of the PLC and we found there was no loss of information. As seen in Figure 3, the accuracy and frequency of the data measured in the cycle are good enough to obtain the required process information.

### 3.2. Variables Measured under Criterion 360

From all the sensors available in the press, we selected the ones that will provide us with the most critical information about the process. The selected sensors are already installed in the press and are used for automatic operation afterwards; they give free information. Table 1 shows the monitored variables.

## 4. Gravity Centre Virtual Sensor

Two conditions must be met to ensure that we are performing correct manufacturing in the stamping process to prevent both excessive wear of the tool die and unfavourable deformations of the press structure. These are:The stamping of the press must be as centred as possible to prevent an imbalance from happening in the upper die of the press. The press is an element that becomes deformed and therefore an excessive effort or an imbalance will favour the appearance of breakdowns and will increase the wear of the different mechanical elements of the equipment.The drawing must be fluid, that is to say, the deformation of the material must be carried out continuously. The lack of fluid drawing can cause two problems; firstly, the material may not be deformed correctly and quality defects may appear, and secondly, this may increase excessive wear between the die and blank surfaces.

Under these conditions, we proceed to obtain the gravity centre of the slide in each degree of cycle position using the classic formula of the calculation of the mass centre, see [25].

In our case, we take the values of the tonnage as masses, which are applied to our physical system defined by the press. Therefore, we can obtain the equations defined below to determine the press imbalance.

$T_{n}^{i}$ is the set of tonnage values for each one of the press positions with a total of 360 data points, where [n]:={1,2,3,4} is the number of sensors and [i]:={0,1,2,⋯,359} is the press position. The calculation of the gravity centre as follows:

We seek to obtain the imbalance in each press position during a cycle; therefore, we obtained the tonnage values of the four sensors are the same degree of press and we replaced them in the same equation. We performed this process in order to obtain the displacement in the plane (X,Y) of each press position. Applying the same equation to both the *X* and *Y* coordinate axes, we obtain:(1)$${x_{G}}_{i}=\frac{T_{1}^{i}a_{1}+T_{2}^{i}a_{2}-T_{3}^{i}a_{3}-T_{4}^{i}a_{4}}{T_{1}^{i}+T_{2}^{i}+T_{3}^{i}+T_{4}^{i}},\;{y_{G}}_{i}=\frac{T_{1}^{i}b_{1}-T_{2}^{i}b_{2}-T_{3}^{i}b_{3}+T_{4}^{i}b_{4}}{T_{1}^{i}+T_{2}^{i}+T_{3}^{i}+T_{4}^{i}}$$
where $a_{1},a_{2},a_{3},a_{4}=a/2$ and $b_{1},b_{2},b_{3},b_{4}=b/2$ correspond to the distances in *X* and *Y* of the tonnage sensor position, respectively, to the centre of the slide, see Figure 5. Being a symmetric system, the equations can be rewritten as:(2)$$P_{i}\left({x_{G}}_{i},{y_{G}}_{i}\right)=\left(\frac{a(T_{1}^{i}+T_{2}^{i}-T_{3}^{i}-T_{4}^{i})}{2(T_{1}^{i}+T_{2}^{i}+T_{3}^{i}+T_{4}^{i})},\;\frac{b(T_{1}^{i}-T_{2}^{i}-T_{3}^{i}+T_{4}^{i})}{2(T_{1}^{i}+T_{2}^{i}+T_{3}^{i}+T_{4}^{i})}\right),\;i=0,\cdots ,359.$$

The vector encompassing the points of the gravity centre of the press in the plane (X,Y) for each press position during a cycle is
(3)$$G=\left[P_{0}\left({x_{G}}_{0},{y_{G}}_{0}\right),\;P_{1}\left({x_{G}}_{1},{y_{G}}_{1}\right),\;\cdots ,\;P_{359}\left({x_{G}}_{359},{y_{G}}_{359}\right)\right].\;$$

Figure 6 shows the values of the gravity centre in a 3D representation within limits that we will call a “pyramid”, where the obtained information can be analysed. The values *X* and *Y* represent the position of the gravity centre, while the coordinate *Z* represents the tonnage value. In orange, we can see the limits set by the manufacturer regarding the recommended imbalance based on the force exerted by the press.

The lower the force, the greater the tolerance to imbalance, and the greater the force, the greater the damage that can produce a minimum imbalance and thus the lower the tolerance. The pyramid defines the maximum limits that the values of the gravity centre on the *X* and *Y* axes can reach for each pressing force, these values are represented on the *Z* axis. In the example shown in Figure 6, the range of forces is from 200 to 2500 tonnes. In summary, what we have is a maximum deviation limit of the gravity centre defined on the *X* and *Y* axes for each force value, superimposing each of the planes obtained for a value of the force on the *Z* axis up to 2500 tonnes. We thus obtain the pyramid shown in the image. Within this area is where we plot the values of the gravity centre obtained.

### Test to Verify the Developed Tool

Tests were carried out to verify that our system actually detected minimal variations in stamping, ensuring its sensitivity and its ability to warn us of any anomalies. Tests were carried out by forcing an imbalance in the stamping with different heights up to 1 mm in one of the four zones of the connecting rod feet. These height differences were defined in various configurations and the system was tested in 16 different slope combinations, forcing the slope knowing the desired curve of the gravity centre to be obtained.

In Figure 7, we can see the set up of the equipment used to carry out the test, where after calibration of the system to ensure that it was perfectly balanced, 1 and 2 mm thick steel plates were placed in various configurations on each of the pillars of the press, forcing the unevenness towards one of the sides of the slide. In Figure 7, a test configuration is shown with the imbalance intentionally forced to the right side of the slide.

In this case, two plates of 1 mm were placed in the pillars on the right, where we expected to see a deviation in the value of the centre of gravity to the right on the X axis and values centred on the Y axis. The result obtained was as expected, as we can see in Figure 8. With all the combinations studied in the test obtaining a result as expected, it can be said that the system worked correctly [26] and therefore it was put into production.

## 5. Gravity Centre Watchdog Alarm (GC-WatchdogAlarm)

This system, called a GC-WatchdogAlarm, monitors the condition of the press stamping and warns us when a pattern appears that we consider outside the correct working standards. To this end, two restrictions have been defined that must not be complied with: one is that there should be no deviation in the stamping and the second is that there should be no difference between each of the force values for each press position greater than usual.

The GC-WatchdogAlarm tool runs every 24 h, analysing all the stamping parts made during a day of production. In the case of finding an anomaly, it generates an alarm that is sent to the maintenance team in order to perform a deep analysis of what happened. This is a good tool to take advantage of when making decisions. Below, we explain each of the restrictions in more detail.

### 5.1. Alarm Limit Calculation

In order to know when the press begins to work with an anomalous behaviour, safety limits have been defined within the pyramid of the press that indicate the degree of malfunction. They have been classified as follows: “under surveillance” behaviour, when it exceeds the defined limit in green; “warning” behaviour, when it exceeds the yellow line; and “urgent revision” behaviour, when values outside the red limit occur. The limit values are calculated as 33% and 66% of the limit value established by the manufacturer. These limits can be seen in the following Figure 9.

As long as the GC value remains within the green zone, this will be considered normal and the alarm sending system will not perform any action after the analysis. When the first limit is exceeded, an alarm is generated to monitor the behaviour of the press. At this point, it is determined whether it is a random anomaly that will not happen again or a repetitive event of strokes outside the normal work area. If this happens, then the person in charge will be notified so that the necessary measures can be taken. When there are values that exceed the yellow limit, the procedure to be followed is the same as just explained, but this time, a thorough inspection of the different elements involved in the stamping process is directly carried out, making the necessary adjustments to return to the normal work situation. Furthermore, finally, if the limit defined in red is exceeded, production must be stopped urgently.

### 5.2. Continuity—DBSCAN

As we have already said, to obtain a good conformation of the piece, the deep drawing process must be continuous, so that the steel sheet slides into the die in a fluid way. The geometry of the design of many parts, together with the thicknesses of the sheets with which we work, need to meet these requirements. Failure to follow these conditions could lead to the occurrence of unfavourable events such as excess friction and wear of the die surface. In addition, the stretching of the material would not be adequate and cracks could appear. Industrial presses are designed to control the speed of the press, where the speed of raising and lowering of the die is higher than that at the time of drawing. In the following Figure 10, one can see the variation in the speed of the press in the degrees that the drawing is carried out, approximately between 100 and 180 degrees.

To detect possible anomalies in this continuous deformation, use of the DBSCAN [27] algorithm is proposed. The DBSCAN algorithm is a non-parametric algorithm of grouping that is designed to form groups of points. DBSCAN requires two parameters: the maximum distance between two points ϵ (eps) and the minimum number of points required to form a dense area/cluster ($min_{p}ts$).

There are very efficient methods to choose the most optimal value of the parameter ϵ, as seen in [28], where a new method is proposed to obtain epsilon based on concepts of computational geometry, In a more specific case [29], they were able to determine the value automatically. In our case, having such a particular pattern and with the distribution of the points similar in all the samples to be analysed, the following process will be followed to obtain the epsilon value that best fits our problem. By monitoring the state of the press at all times, we have access to a record of data of thousands of cycles. Once verified that the information to be analysed has been stored under normal working conditions of the press, the maximum distances of each cycle between two consecutive points of the centre of gravity are obtained so we can subsequently determine the mean of the calculated distances. This value will be the one that we will use as ϵ when applying the DBSCAN algorithm to our data to analyse the press state.

The value of the minimum number of points, $min_{p}ts$, of a given point so that it can or cannot be classified as a core point will be 2, as this is the same number considered within the min_samples in the DBSCAN algorithm. This ensures that at the moment that more than one point appears at a distance greater than what is considered normal for the stamping of the analysed part, the system notifies us by sending an alarm.

It should be noted that unlike the centred stamping analysis that is performed for all dies equally, in this case, DBSCAN will be applied differently to each of the dies used in the press. This is due to the geometry that each piece has, where the resulting force obtained will have a different behaviour depending on the drawing distance, some areas with greater depth than others or the adjustment of the die limits, among others. For these reasons, the value of ϵ will be different for each die. So the Algorithm 1 will be used for each car part with its specific data input.
**Algorithm 1:** Abnormal Continuity Detection Algorithm**Data**: gcData: Gravity centre data of a press cycle to analyse**Data**: gcHist: History of gravity centre data**Data**: minSamp: Minimum sample for DBSCAN algorithmeps←0;        /*(Initialise epsilon value( */epsAvg←array[];       /*(Initialise empty array( */
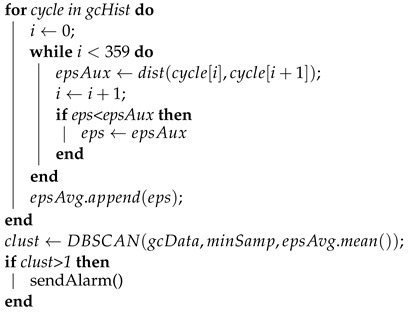


DBSCAN was adapted and the distances were taken between points in three dimensions to obtain the number of clusters. However, if we want to obtain the epsilon of the record, and subsequently apply the algorithmic DBSCAN, this will result in a very high computational cost. To reduce the calculation time, the alternative was obtaining the number of clusters in each cycle twice in 2D, one for the data in the X−Z plane and the other for the Y−Z plane. This is much more optimal in terms of computational cost.

## 6. Results

Once the proposed method was tested and it was verified that it works correctly, it was put into production. The I3oT tool was used to monitor the balance of the press in the facilities of the Ford press workshop in Valencia. The press model is FAGOR LE4-2500-4600-2500 and is controlled by a PLC Siemens CPU 317-2 DP. This press manufactures a large number of parts of different vehicles at an average speed of 16 HPM (hits per minute). The tool was introduced a year ago and since then three anomalies have been detected, which are described below.

### 6.1. Anomaly 1

The first identified case shown in Figure 11 was detected with the die of the roof of a Ford Transit van, where the alarm system informed us that the value of the tonnage was around 2300 Tm, while the theoretical value defined for this item should be around 1760 Tm. The tonnage was above the safety limit value of the top of the pyramid, 2000 Tm. In addition, it far exceeded the theoretical value defined in the design phase, which led to excess effort of the press and the problems that this entails. After finding out the root of the problem, tests were carried out with the equipment to reduce the press effort without affecting the quality of the part. Finally, the working parameters of the press were adjusted. In this case, the slide adjustment was modified and the tonnage was reduced by up to 700 tons, therefore leaving the press working at 1600 Tm without affecting the quality of the part and working even below the design value obtained from the simulation.

### 6.2. Anomaly 2

The second found case shown in Figure 12 had the same pathologies as explained in anomaly 1. The body side part of the Galaxy was being manufactured at around 2400 tons, while the theoretical tonnage value obtained from the simulation was specified at 1639 Tm. After completing different tests, the force of the press was reduced without affecting the quality of the part. This was solved by modifying the adjustment of the slide in order to reduce the tonnage to the theoretical values defined in the design phase. In the two cases explained, it was concluded that the problem appeared due to a bad configuration of the work parameters of the press.

### 6.3. Anomaly 3

The last case of an anomaly detected in the press was due to an alarm sent through the GC-WatchdogAlarm system using the DBSCAN algorithm that identified an item that was working without the desired continuity in the drawing and giving a result of two identified clusters, as clearly seen in Figure 13. In this case, the problem was the outer door panel part of a Ford S-Max. The maintenance workers, when they went to check the press, found that it emitted unusual sounds and found stronger vibrations than usual in the concrete structure on which the press was mounted. This did not happen continuously but occurred at certain random stamping strokes. The workers decided to use the web application to consult the stored history of the work parameters and analyse it to obtain new information that could help to understand what was happening, since on previous occasions revisions had been made during the manufacture of this part without finding any anomaly in the press. The data were analysed in the web application and it was verified that the stored tonnage curves did not have the usual behaviour, since during drawing, the value of the tonnage should gradually increase, but in this case it had not happened. In the image shown below, we have a normal tonnage value on the left and the die curve of the S-Max outer door panels on the right.

In the bottom curve, marked in green we can see how the tonnage gradually increases during deep drawing. We can say that continuous drawing is being carried out; this being the right process during the stamping of a part. The opposite happens with the part that is presenting problems during manufacture, since we can observe discontinuity during drawing, which indicates that something is not working properly.

After a thorough analysis of the configuration of the press, it was found that the punch of the die was slightly higher than the blank holder, and this prevented drawing from being fluid. For this reason, one could feel the strokes that transmitted great vibrations to the entire press. After modifying the cushion configuration to change the blankholder and punch height, the problem was solved and excessive vibrations were avoided. In Figure 14, you can see the gap between the values of the centre of gravity within the pyramid.

### 6.4. Results Summary

To summarise, we show the results obtained in Table 2, where we can see the amount of effort reduced thanks to the methodology presented in this paper.

In the first two anomalies, the value of the effort made by the press was greatly reduced. In the manufacture of the roof part of the vehicle, we have reduced the tonnage from 2300 to 1600 Tm, reducing it by about 700 Tm. We have done the same in the manufacturing part of the side part of the body, where the tonnage was reduced by 800 Tm. Considering the press as an elastic element that can be deformed, we can confirm that the force was reduced both at the structural level and in the different mechanical elements. This allows us to extend the useful life of the equipment and avoid unexpected breakdowns in the short term. In addition, the energy consumption was reduced, since the effort made is lower. In the third anomaly detected, where two clusters were found in the data of the calculated gravity centre, we can confirm that the wear was reduced both on the die surface and on the blank surface. In addition, we reduced the vibrations at the structural level that can be harmful to the press.

## 7. Conclusions

The use of existing data to develop a new application is an important concept for the development of Industry 4.0. Many companies and research teams develop their algorithms and ideas from the installation of new sensors without taking into account the implications that these proposals have in real factories. The cost of the sensor, installation setup, maintenance, machine modifications and replicating this for hundreds of machines causes many of the ideas to fail when the innovation is implemented in industry. Therefore, in this article, and in previous work of the group [19,20], a novel concept is presented for Industry 4.0 for the development of algorithms by exclusively using the information that is available in the machines. In [24], this philosophy is defined as I3oT (industrialisable industrial Internet of Things), since through the use of what is already installed, tools can be generated with a very low impact and this can generate great benefits. The machines, as well as the presses and in general the automated systems, are full of sensors for their normal automated production. These sensors are installed and the information they provide is almost free. Following this philosophy, in this article, the Criterion-360 was developed in order to measure many variables, not only tonnage, but also variables such as the press speed, the counterweight and overload pressure, the cushion pressure and position, lubrication, etc. Therefore, many possibilities arise when this amount of information is used. One of them is the proposal of this tool that allows us to detect unusual behaviour in real time. In the present work, the press balance is measured by a virtual sensor that, by using real-time tonnage information, allows calculating the gravity centre of a virtual load that moves the mould up and down. The imbalance watchdog checks two parameters: the imbalance itself and discontinuity in the press effort. The DBSCAN algorithm is used to check the second parameter. Our future works will focus on implementing the C360 as a standard at Ford Motor Company in order to monitor all presses at the plant in Valencia, as well as at other plants. As the tool is based on the I3oT philosophy, the cost of expanding the tool is low. The team is currently working on new tools and algorithms to use with C-360. We are developing a novel algorithm to adjust the parameters in order to save energy and also developing a digital twin for determining the quality of the parts.

## Figures and Tables

**Figure 1 sensors-23-06569-f001:**
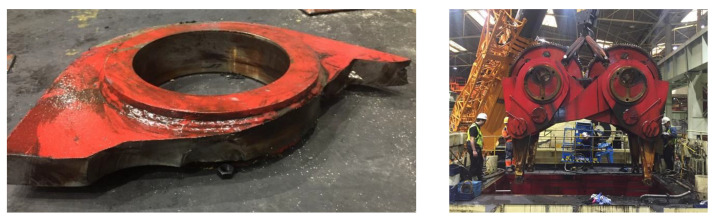
Kinematic chain breakdown of a mechanical press.

**Figure 2 sensors-23-06569-f002:**
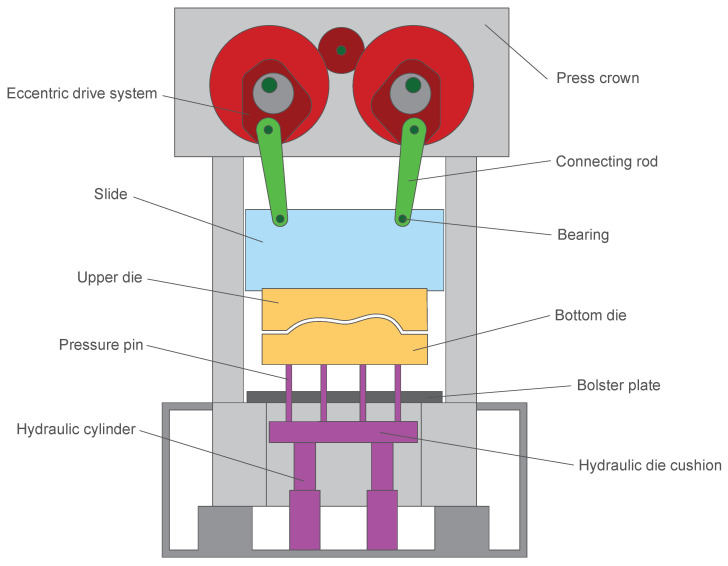
Single-action mechanical press sketch.

**Figure 3 sensors-23-06569-f003:**
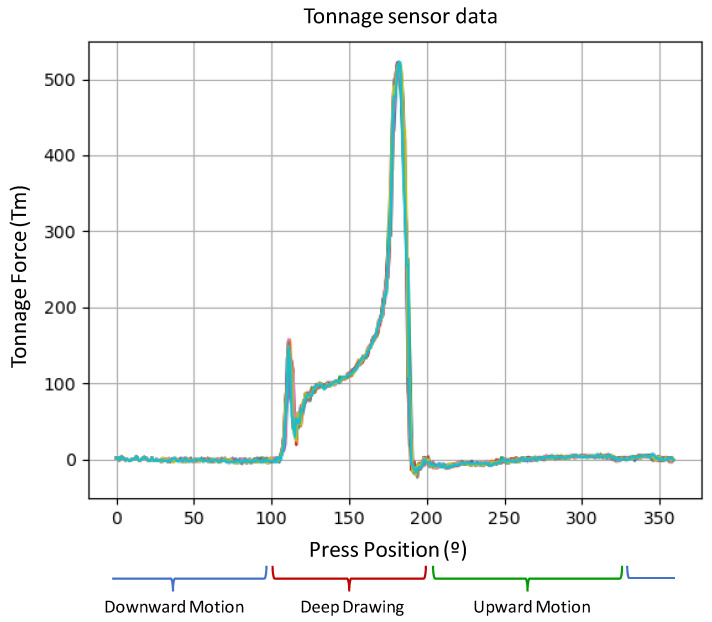
Tonnage data of a press cycle.

**Figure 4 sensors-23-06569-f004:**
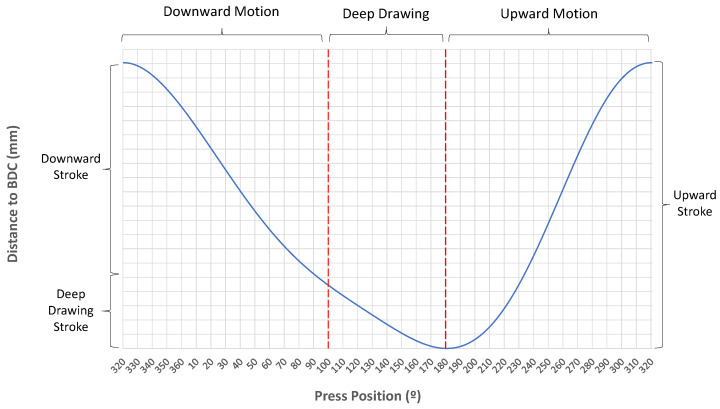
Press position vs. slide distance to BDC. Detailed *Y* axis information is not shown due to confidentiality.

**Figure 5 sensors-23-06569-f005:**
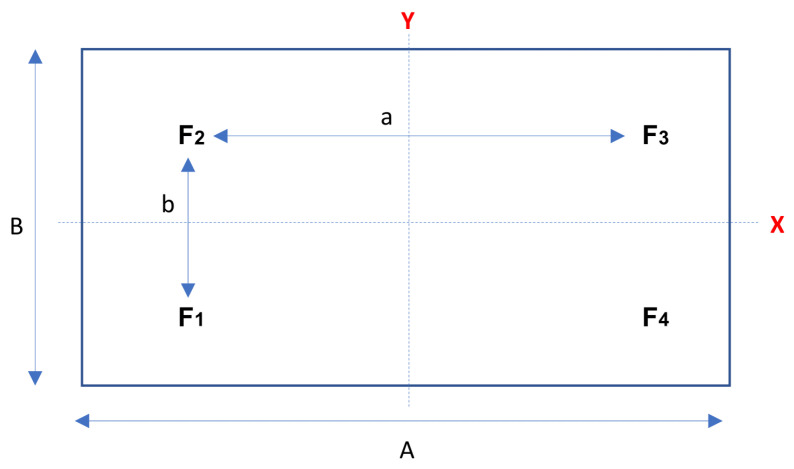
Sketch of the press slide top view with the four connecting rod feet positions. A,B are slide measurements. And a,b are distance between rod feet.

**Figure 6 sensors-23-06569-f006:**
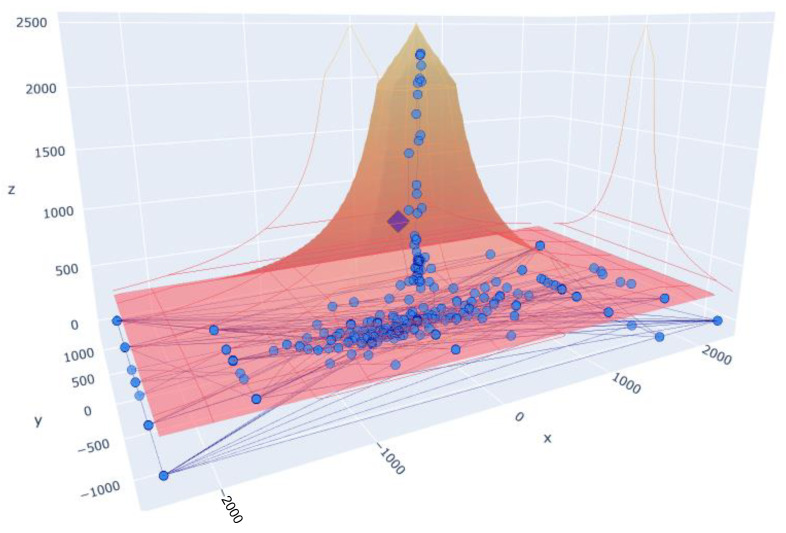
Gravity centre data of a stamping cycle.

**Figure 7 sensors-23-06569-f007:**
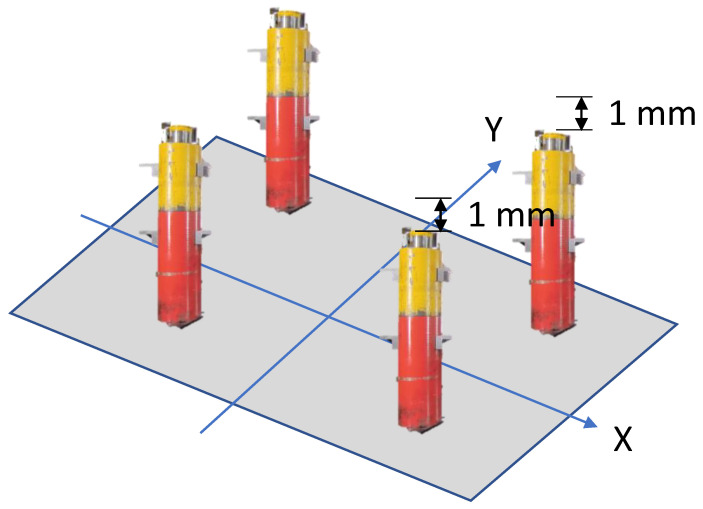
Set up equipment with an imbalance of 1 mm on the right side of the press.

**Figure 8 sensors-23-06569-f008:**
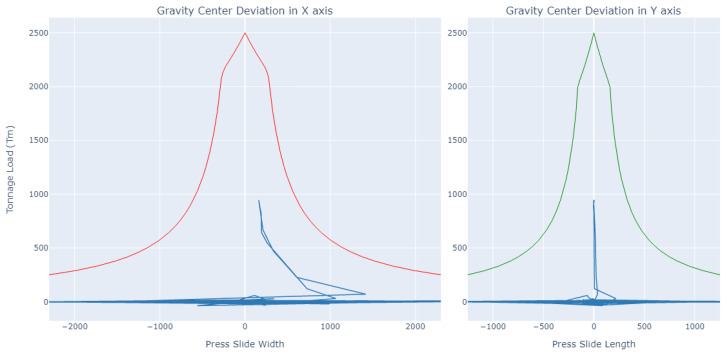
Imbalance test results obtained with the GC tool. Red and green lines are press working limits. Blue lines are gravity center measurements.

**Figure 9 sensors-23-06569-f009:**
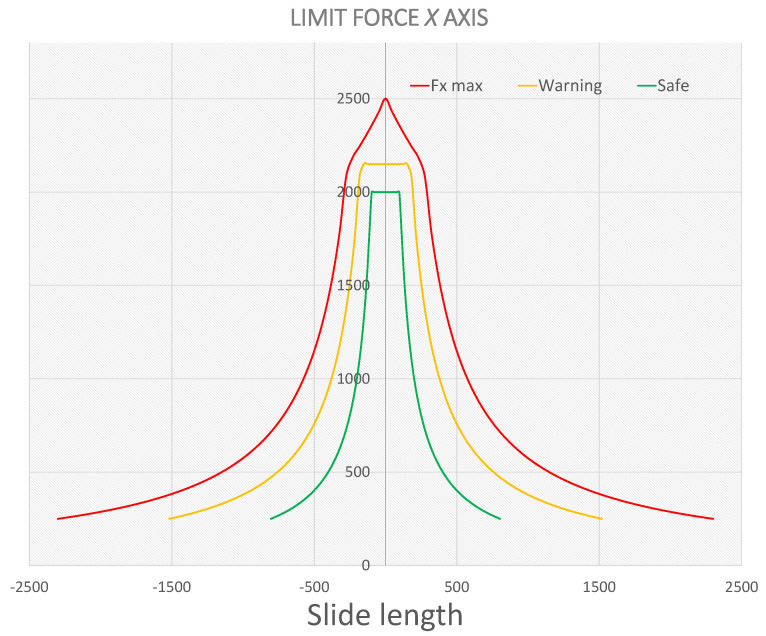
Gravity centre safety limits.

**Figure 10 sensors-23-06569-f010:**
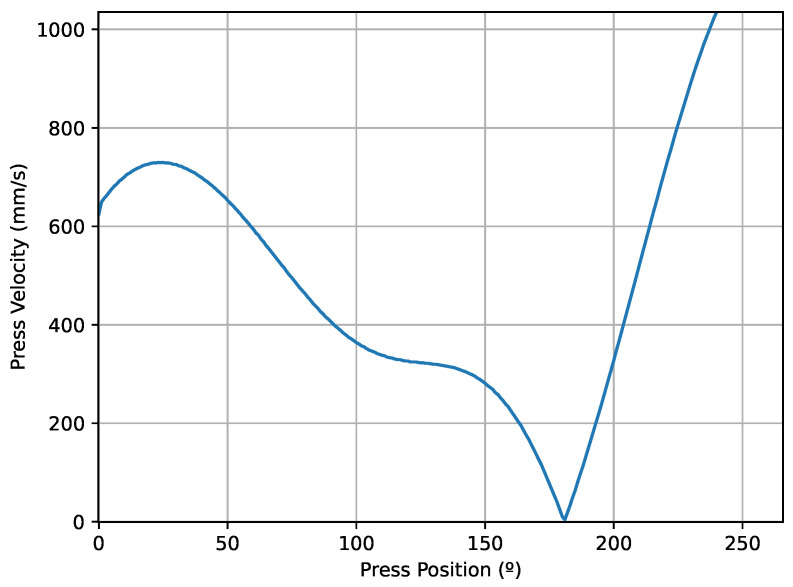
Press velocity during a cycle.

**Figure 11 sensors-23-06569-f011:**
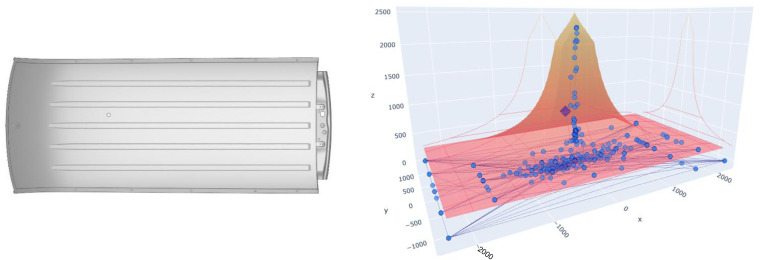
Roof part. Tonnage values exceeding top working limits.

**Figure 12 sensors-23-06569-f012:**
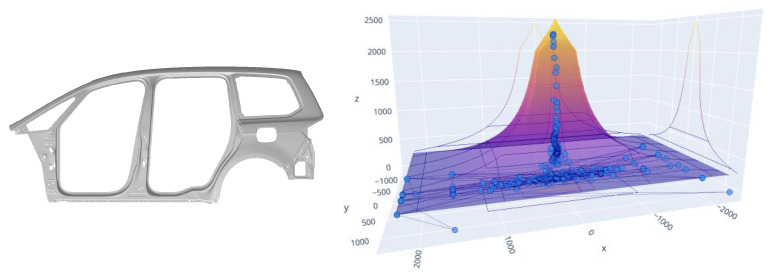
Body side part. Tonnage values exceeding top working limits.

**Figure 13 sensors-23-06569-f013:**
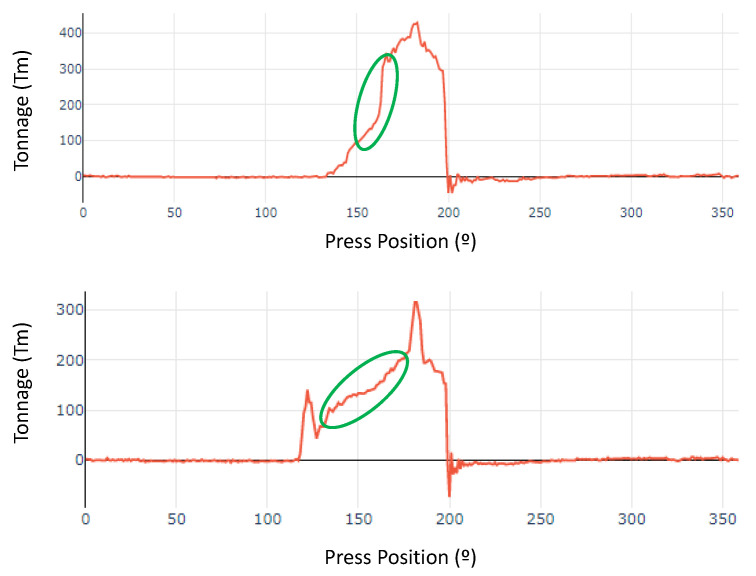
Outer door panels. Abnormal tonnage value (**top**). Correct tonnage value (**bottom**). Green circle indicates load during deep drawing.

**Figure 14 sensors-23-06569-f014:**
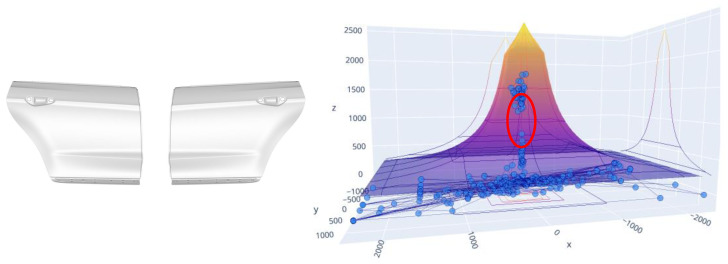
Outer door panels. Gravity centre alarm without continuity. Red circle indicates discontinuity during deep drawing.

**Table 1 sensors-23-06569-t001:** Monitored variables in a stamping press.

Process Value	Sensor Quantity	Units	Criterion
Tonnage Force	4	Tm	360
Cylinder cushion pressure	8	bar	360
Cylinder cushion pressure	2	mm	360
Counterbalance pressure	1	bar	360
Overload pressure	2	bar	360
Energy consumption	1	W/h	360
Clutch activation time	1	ms	Mini-Term
Brake activation time	1	ms	Mini-Term
Slide position	1	mm	Mini-Term
Press speed	1	hit/min	Mini-Term
Die clamp travelling time	12	ms	Mini-Term
Lubrication mapping	1	l/s	-

**Table 2 sensors-23-06569-t002:** Results of parts identified as anomalies.

Car Part	Prev. Tonnage	Post Tonnage	Clusters	Safety Issue
Roof	2327	1618	1	Exceed warning limit
Body side	2415	1625	1	Exceed warning limit
Outer door panels	1720	1720	2	Continuity

## Data Availability

Not applicable.

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
