# Peer review of "Virtual Sensor of Gravity Centres for Real-Time Condition Monitoring of an Industrial Stamping Press in the Automotive Industry"

_sensors, 2023, doi:10.3390/s23146569_

Round 1

Reviewer 1 Report

The content of the paper has little to do with Industrial Sensors journal, as it does not deal with them at all, but rather with a journal of Mechanical Technology like Manufacturing Systems Journal or similar.

The topic dealt with concerns the monitoring of the correct functioning of a mechanical press, in the logic of Industry 4.0, to prevent malfunctions that may compromise the printed products but especially the integrity of the machine, whose extraordinary maintenance involves very high costs.

The paper is certainly interesting from the technological process point of view, although it must be reviewed in the organization of the sections and from the linguistic point of view, not always fluid and clear in reading. English is sometimes incorrect (example, line 169). For example, the introduction should be completely revised and perhaps supplemented with parts of paragraph 2, especially the initial part of section 2 (lines 57-70); similarly, subparagraphs from line 137 to line 154 may be the conclusion of the introduction.

The introduction has to report the state of the art of the theme addressed and then to highlight the innovative contribution to be provided and how it is intended to develop it.

Long phrases, such as from line 72 to line 78, are heavy and not easy to read and understand.

The figures must be placed downstream of their quotation and not before it. Figure 10 is never mentioned in the text.

It is not a paper to be published in Industrial Sensors J., in my opinion, even though it may be worthy of

publication in a journal.

English is fine, but it should be checked where indicated, as it seems to me that a verb is missing in that phrase.

Author Response

Thank you for taking the time to review our paper. We hope that this new version of the manuscript Virtual Sensor of Gravity Centre for Real Time Condition Monitoring of an Industrial Stamping Press in Automotive Industry meets the necessary standards for publication in the Sensors journal.

What follows is a record of the changes made to the initial submission of the paper attending to the comments of the reviewers. Some paragraphs have been moved, some added and some modified. Additions to the paper from the initial submission are highlighted in red in the new PDF File.  Below, the suggestions of the reviewers are considered and answered in turn.  

Finally, some minor changes have been carried out in order to maintain the logic of the paper wording and to correct some typographic errors identified during the revision of the paper. 

Suggested modification 1: “The content of the paper has little to do with Industrial Sensors journal, as it does not deal with them at all, but rather with a journal of Mechanical Technology like Manufacturing Systems Journal or similar. The topic dealt with concerns the monitoring of the correct functioning of a mechanical press, in the logic of Industry 4.0, to prevent malfunctions that may compromise the printed products but especially the integrity of the machine, whose extraordinary maintenance involves very high costs.”

Response to the reviewer:  Dear reviewer, thank you for your comments. We perfectly understand your reflection. However, we believe that the main point of the present paper is how to develop a virtual sensor that computes the center of gravity of a virtual load that moves the slice upwards and downwards using the physicals sensors data installed in the press. It is true that the application is to monitoring of the correct functioning of a mechanical press and it could be publish in Manufacturing Systems Journal or similar but the topic of the special issue in the “Sensors Journal” is “Virtual Sensors for the Industry 4.0 Era” and we strongly believe that fix perfectly the main topic of the paper.

Suggested modification 2: “The paper is certainly interesting from the technological process point of view, although it must be reviewed in the organization of the sections and from the linguistic point of view, not always fluid and clear in reading. English is sometimes incorrect (example, line 169). For example, the introduction should be completely revised and perhaps supplemented with parts of paragraph 2, especially the initial part of section 2 (lines 57-70); similarly, subparagraphs from line 137 to line 154 may be the conclusion of the introduction.”

Response to the reviewer:   Done. We agree with the reviewer and these paragraphs are moved to the introduction. A deep rewriting of the introduction was done. We hope the new version fits the reviewer exigence.

Suggested modification 3: The introduction has to report the state of the art of the theme addressed and then to highlight the innovative contribution to be provided and how it is intended to develop it.”

Response to the reviewer:  Done

Suggested modification 4: Long phrases, such as from line 72 to line 78, are heavy and not easy to read and understand.”

Response to the reviewer:  A deep rewriting of the introduction was done. We hope the new version fits the reviewer exigence.

 Suggested modification 5: The figures must be placed downstream of their quotation and not before it. Figure 10 is never mentioned in the text.”

Response to the reviewer:  Done. Figures placed after their quotation. Figure 10 ref remarked in red at the end of section Results.

 Suggested modification 6: “It is not a paper to be published in Industrial Sensors J., in my opinion, even though it may be worthy of publication in a journal.“

Response to the reviewer:  Authors believe that the main point of the present paper is how to develop a virtual sensor that computes the center of gravity of a virtual load that moves the slice upwards and downwards using the sensors installed in the press. It is true that the application is to be monitoring of the correct functioning of a mechanical press and it could be publish in Manufacturing Systems Journal or similar but the topic of the special issue in the “Sensors Journal” is “Virtual Sensors for the Industry 4.0 Era” and we strongly believe that fix perfectly the main topic of the paper.

Reviewer 2 Report

In the section  8.2 Anomaly 2, the autors mentioned:

………Press configuration parameters were adjusted and the press force was reduced to standard working values.

there is no mention that standard is used to validate the process.

In the sections 5. According to the new criteria, what are they based on so that only 360 pieces of data are obtained? Are these data enough? Justify your answer:

…………….This new criterion creates specific measures for the press since it allows us to obtain 360 164

data of a variable chosen from the process for a work cycle.

Compared with other methods, what has been the error of the proposed method? What makes the proposed method better than others? It is not mentioned in the paper.

There is no comparative table of the existing methods, advantages and disadvantages must be mentioned.

A minor revision of the English writing is required, review in detail.

Author Response

Thank you for taking the time to review our paper. We hope that this new version of the manuscript Virtual Sensor of Gravity Centre for Real Time Condition Monitoring of an Industrial Stamping Press in Automotive Industry meets the necessary standards for publication in the Sensors journal.

What follows is a record of the changes made to the initial submission of the paper attending to the comments of the reviewers. Some paragraphs have been moved, some added and some modified. Additions to the paper from the initial submission are highlighted in red in the new PDF File.  Below, the suggestions of the reviewers are considered and answered in turn.  

Finally, some minor changes have been carried out in order to maintain the logic of the paper wording and to correct some typographic errors identified during the revision of the paper. 

Suggested modification 1: “In the section 8.2 Anomaly 2, the authors mentioned: ………Press configuration parameters were adjusted and the press force was reduced to standard working values.

There is no mention that standard is used to validate the process.”

Response to the reviewer:  Done. ‘Standard working values’ was referred to tonnage values obtained from the simulation in the previous step of designing process. Modification in the text to make it clear.

Suggested modification 2: “In the sections 5. According to the new criteria, what are they based on so that only 360 pieces of data are obtained? Are these data enough? Justify your answer:

…………….This new criterion creates specific measures for the press since it allows us to obtain 360 data of a variable chosen from the process for a work cycle.“

Response to the reviewer:  Done, justification added in section 3.

Suggested modification 3: “Compared with other methods, what has been the error of the proposed method? What makes the proposed method better than others? It is not mentioned in the paper. “There is no comparative table of the existing methods, advantages and disadvantages must be mentioned.”

Response to the reviewer:  Done. We introduce some new paragraphs to justify the value of the proposal comparing with existing ones and with our previous works. Also the goal of our research is rewritten to clarify it.

Reviewer 3 Report

Although the issue raised in the manuscript is important, the paper contains many inaccuracies and should be corrected.

There is not enough text about related works. Authors don't compare their method to any others. It is unclear what the authors' original contributions are.

The mathematical descriptions should be corrected and the algorithm presented in the manuscript is not described precisely enough. 

The data used for experiments should be described in more detail (more extensively).  

Below there are detailed comments:

line 142, "through the use of these data" it is not clear what data

line 149, "In section 3 we show" the section's numbers described in this paragraph are incorrect

line 170, the content of tab.1 should be described more precisely, the sentence "The following table shows the monitored variables" is not enough

line 185, it should be added that n is related to the number of sensors for processing tonnage force

I have doubts about equations (1) - (3). The parameters a1-a4 and b1-b4 are not visible in Figure 2. I assume that these parameters are positive values that's why we see minuses in formulas but as far as you can understand the location of the minuses for x_Gi, and for y_Gi they are incomprehensible to me. In my opinion, it should be minuses before (2 and 3) or (1 and 4). Moreover why sum in the denominator doesn't have a summation index? I assume it is n (not i). If I'm right, in the case of the sum of four elements, the use of the generalized sum sign seems unnecessary.

In the case of the formula (2) I am wondering about the number 2 in the denominator. Is it means that a1=a2=a3=a4=a/2? The same for b?

Why in formula (3) the last element is Pi(xGi, yGi) - the index is i (not 359)? I wonder why in (2) Pi is not described as Pi(xGi, yGi).

I suggest dividing Fig. 4  into two parts (separate figures Fig. 4a and 4b) and the description of the figure will be able to be more precise.

line 316, the sentence there seems redundant

I think that the content of Fig. 7 or Fig. 8 should be described more extensively

lines 336, 348, etc. "In the image shown below", "In the following image" references to figures should be made using their numbers not positions in the text.

Author Response

Although the issue raised in the manuscript is important, the paper contains many inaccuracies and should be corrected.

There is not enough text about related works. Authors don't compare their method to any others. It is unclear what the authors' original contributions are.

The mathematical descriptions should be corrected and the algorithm presented in the manuscript is not described precisely enough. 

The data used for experiments should be described in more detail (more extensively).  

Thank you for taking the time to review our paper. We hope that this new version of the manuscript Virtual Sensor of Gravity Centre for Real Time Condition Monitoring of an Industrial Stamping Press in Automotive Industry meets the necessary standards for publication in the Sensors journal.

What follows is a record of the changes made to the initial submission of the paper attending to the comments of the reviewers. Some paragraphs have been moved, some added and some modified. Additions to the paper from the initial submission are highlighted in red in the new PDF File.  Below, the suggestions of the reviewers are considered and answered in turn.  

Finally, some minor changes have been carried out in order to maintain the logic of the paper wording and to correct some typographic errors identified during the revision of the paper. 

Suggested modification 1: “line 142, "through the use of these data" it is not clear what data”

Response to the reviewer:  We modify the text to clarify this issue

Suggested modification 2: “line 149, "In section 3 we show" the section's numbers described in this paragraph are incorrect “

Response to the reviewer:  Done

Suggested modification 3: “line 170, the content of tab.1 should be described more precisely, the sentence "The following table shows the monitored variables" is not enough “

Response to the reviewer:  Done

Suggested modification 4: “line 185, it should be added that n is related to the number of sensors for processing tonnage force”

Response to the reviewer:  Done

Suggested modification 5: “I have doubts about equations (1) - (3). The parameters a1-a4 and b1-b4 are not visible in Figure 2. I assume that these parameters are positive values that's why we see minuses in formulas but as far as you can understand the location of the minuses for x_Gi, and for y_Gi they are incomprehensible to me. In my opinion, it should be minuses before (2 and 3) or (1 and 4). Moreover, why sum in the denominator doesn't have a summation index? I assume it is n (not i). If I'm right, in the case of the sum of four elements, the use of the generalized sum sign seems unnecessary.”

Response to the reviewer:  Equations are modified to solve the issues

Suggested modification 6: “In the case of the formula (2) I am wondering about the number 2 in the denominator. Is it means that a1=a2=a3=a4=a/2? The same for b?”

Response to the reviewer:   It is as you mention, modification done.

Suggested modification 7: “Why in formula (3) the last element is Pi(xGi, yGi) - the index is i (not 359)? I wonder why in (2) Pi is not described as Pi(xGi, yGi).”

Response to the reviewer:   Modifications done to define the equations accordingly.

Suggested modification 8: “I suggest dividing Fig. 4  into two parts (separate figures Fig. 4a and 4b) and the description of the figure will be able to be more precise.”

Response to the reviewer:  Thank you for your comment. We noticed that the axis labels are illegible, and we decide to split the figure in two figures. We hope that this modification fits your expectations.

Suggested modification 9: “line 316, the sentence there seems redundant”

Response to the reviewer:  Solved

Suggested modification 10: “I think that the content of Fig. 7 or Fig. 8 should be described more extensively”

Response to the reviewer:  A more detailed explanation has been added.

Suggested modification 11: “lines 336, 348, etc. "In the image shown below", "In the following image" references to figures should be made using their numbers not positions in the text.”

Response to the reviewer:  Done

Reviewer 4 Report

The paper presents a virtual sensor approach to measure the Center of Gravity (CoG) of a sheet metal stamping press. I suggest that the work can be considered for publication after the authors address a few points:

1. lines 50 to 85: I suggest adding a diagram of the mechanism, to help readers not familiar with this technology understand the object of the study.

2. line 102: "... high economic impact for the company, which can be over half a million euros" I suggest to reference a source for this figure, or to indicate only an order of magnitude.

3. lines 7 and 164: the authors claim that the proposed strategy permits to obtain "360 data" or "5400 data" of the measured variables. However, this statement is unclear at this point. I assume that the algorithm samples a sensor output 360 times in a cycle of the press, but please clarify the description of the proposed strategy. In addition, the sampling strategy is not specified: are the 360 measurements uniformly distributed in time? or in the angular position of a rotating element of the mechanism (which may not rotate at constant speed)?

4. Section 6: The authors shall point out why the measurement of CoG is important to detect failures of the press or defects in the stamping process. E.g. how does a failure affect the CoG position of the press?

5. line 185: what is the meaning of subscripts n and i? I assume that n identifies the sensor and i identifies the measurements, but the sampling of measurements is unclear (see also comment 3.).

6. Section 7.2: Please provide a brief description of the DBSCAN algorithm. The acronym appears here for the first time and is not defined, and the method is not properly described.

The quality of English Language is mostly acceptable, but some statements are unclear (see for example comment 3 in the section "Comments and Suggestions for Authors
" of this revision form.

I suggest that a moderate editing of English language is needed to improve the readability of the paper.

Author Response

The paper presents a virtual sensor approach to measure the Center of Gravity (CoG) of a sheet metal stamping press. I suggest that the work can be considered for publication after the authors address a few points:

Thank you for taking the time to review our paper. We hope that this new version of the manuscript Virtual Sensor of Gravity Centre for Real Time Condition Monitoring of an Industrial Stamping Press in Automotive Industry meets the necessary standards for publication in the Sensors journal.

What follows is a record of the changes made to the initial submission of the paper attending to the comments of the reviewers. Some paragraphs have been moved, some added and some modified. Additions to the paper from the initial submission are highlighted in red in the new PDF File.  Below, the suggestions of the reviewers are considered and answered in turn.  

Finally, some minor changes have been carried out in order to maintain the logic of the paper wording and to correct some typographic errors identified during the revision of the paper. 

Suggested modification 1: “lines 50 to 85: I suggest adding a diagram of the mechanism, to help readers not familiar with this technology understand the object of the study.”

Response to the reviewer:  Press diagram added.

Suggested modification 2: “line 102: "... high economic impact for the company, which can be over half a million euros" I suggest to reference a source for this figure, or to indicate only an order of magnitude. “

Response to the reviewer:  Solved

Suggested modification 3: “lines 7 and 164: the authors claim that the proposed strategy permits to obtain "360 data" or "5400 data" of the measured variables. However, this statement is unclear at this point. I assume that the algorithm samples a sensor output 360 times in a cycle of the press, but please clarify the description of the proposed strategy. In addition, the sampling strategy is not specified: are the 360 measurements uniformly distributed in time? or in the angular position of a rotating element of the mechanism (which may not rotate at constant speed)?

Response to the reviewer:  The issue its been clarifyed in the text, we hope it makes it clear now.

Suggested modification 4: “Section 6: The authors shall point out why the measurement of CoG is important to detect failures of the press or defects in the stamping process. E.g. how does a failure affect the CoG position of the press?”

Response to the reviewer:  Done

Suggested modification 5: “line 185: what is the meaning of subscripts n and i? I assume that n identifies the sensor and i identifies the measurements, but the sampling of measurements is unclear (see also comment 3.).”

Response to the reviewer:  Added.

Suggested modification 6: “Section 7.2: Please provide a brief description of the DBSCAN algorithm. The acronym appears here for the first time and is not defined, and the method is not properly described.”

Response to the reviewer:  Description added, I hope it fits your expectations.

Reviewer 5 Report

1. The abstract lacks the highlighting contribution and novelty clearly.

2. The literature is weak and it has to be enriched by other relevant works. According to drawbacks of these works, the authors have to draw the motivation behind this study.  

3. The introduction has to be enriched with other new prediction works. I suggest the following relevant reference: doi.org/10.3390/pr11051507.

4. There is general introduction in the manuscript has to be reduced.

5. The schematic block diagram has to be presented in the introduction part.

6. The authors have to explain the meaning of "criterion" in Table (1).

7.  The two conditions in section 6 has to be clearly explained.

8. The authors have to explain if the method of center calculation belongs to them or it has been inspired from other study.

9. Figure 9 has low resolution and the definition of axes are required.

10. The authors did not displayed specimen based on experimental implementation.

11. The authors have to conduct a comparison study with previous works.

12. The study has not been supported by numeric evaluation or Tables.

13. The results are few and it has to be enriched by other verification of proposed method.

14. The doors in Figure 10 refers to what?!!

15. The conclusion is descriptive and it does not reflect the acquired results.

16. The future work has to be added. 

Minor edit of English language is required

Author Response

Thank you for taking the time to review our paper. We hope that this new version of the manuscript Virtual Sensor of Gravity Centre for Real Time Condition Monitoring of an Industrial Stamping Press in Automotive Industry meets the necessary standards for publication in the Sensors journal.

What follows is a record of the changes made to the initial submission of the paper attending to the comments of the reviewers. Some paragraphs have been moved, some added and some modified. Additions to the paper from the initial submission are highlighted in red in the new PDF File.  Below, the suggestions of the reviewers are considered and answered in turn.  

Finally, some minor changes have been carried out in order to maintain the logic of the paper wording and to correct some typographic errors identified during the revision of the paper. 

Suggested modification 1: “The abstract lacks the highlighting contribution and novelty clearly.”

Response to the reviewer:  Abstract is rewriten

Suggested modification 2: “The literature is weak and it has to be enriched by other relevant works. According to drawbacks of these works, the authors have to draw the motivation behind this study.“

Response to the reviewer:  Done. We hope the new version fits the reviewer exigence.

Suggested modification 3: “The introduction has to be enriched with other new prediction works. I suggest the following relevant reference: doi.org/10.3390/pr11051507. “

Response to the reviewer:  Done. Thaks for the reference recomendation

Suggested modification 4: “There is general introduction in the manuscript has to be reduced.”

Response to the reviewer:  Done. We hope the new version fits the reviewer exigence.

Suggested modification 5: “The schematic block diagram has to be presented in the introduction part.”

Response to the reviewer:  Sorry but here we don't know which block diagram you mean. 

Suggested modification 6: “The authors have to explain the meaning of "criterion" in Table (1).”

Response to the reviewer:  Explanation done

Suggested modification 7: “The two conditions in section 6 has to be clearly explained.”

Response to the reviewer:  Done

Suggested modification 8: “The authors have to explain if the method of center calculation belongs to them or it has been inspired from other study.“

Response to the reviewer:  We modify the introduction and the goal of our research to clarify how our previous wors inspire the present study

Suggested modification 9: “Figure 9 has low resolution and the definition of axes are required.“

Response to the reviewer:  Fixed

Suggested modification 10: “The authors did not displayed specimen based on experimental implementation.”

Response to the reviewer:  Done. This issue is explained at the start of the result section.

Suggested modification 11: “The authors have to conduct a comparison study with previous works.”

Response to the reviewer:  Done

Suggested modification 12: “The study has not been supported by numeric evaluation or Tables.”

Response to the reviewer:  We include a subsection at the end of the results as a summary explaining the benefits, including a table to compare the improvement.

Suggested modification 13: “The results are few and it has to be enriched by other verification of proposed method.”

Response to the reviewer:  Thank you for your comment. As we explained in the introduction the fault in press shops is not common. The results of the paper are obtaining awaiting the failure of the press during one year. For that reason, we made a test, that is 4.1 point to force the inbalance and to validate the tool previously to connect it with the real production.

Suggested modification 14: “The doors in Figure 10 refers to what?!!”

Response to the reviewer:  Fixed

Suggested modification 15: “The conclusion is descriptive and it does not reflect the acquired results.”

Response to the reviewer:  The conclusion are improved

Suggested modification 16: “The future work has to be added.”

Response to the reviewer:  Future works are added at the end of the paper

Round 2

Reviewer 1 Report

I welcome the authors' justifications. However, corrections remain to be made.

1. You have to reverse figures 3 and 4: 4 cannot be recalled and shown before the 3!
2. there are language errors and typos still in the text (e.g., line 36, 83, etc) that need to be corrected. I advice to review the text as a whole: e.g., adverbs of place or time can not precede the object complement, usually, and they preferably go at the end of the phrase.

I welcome the authors' justifications, but draw up some corrections:
1. Figures 3 and 4 must be reversed: the 4 cannot be recalled and shown before the 3!
2. there are several language errors and typos (e.g., line 36, 83, etc.). Again, adverbs of place or time cannot precede the object complement, normally, but generally go to the end of the sentence.

Author Response

Thanks for your suggestions. 

Figures have been swapped.

English has been revised and grammatical errors corrected. 

Reviewer 3 Report

The text of the manuscript has been significantly improved. However, there are many minor grammatical errors in the text and from time to time the style of the issues described is too casual.

The following two mistakes must be corrected.

lines 203-204 I can read "See deep drawing stage, the stroke it is between 150-200 mm depending of the die. Here we are reading between 100-80 data values, around 1 data value each 2mm of the stroke." but I have no idea where I can confront the data. Perhaps it is due to the figure 3, but there is no vertical axis described there (no units).

line 413 "As summary we want to show the results obtained in the tbale ?? where" there is no table number.

In my opinion, there are many small mistakes that make reading comprehension difficult, e.g. line 90, "But still existing challenges to overtake in predictive maintenance." And sometimes the sentences are too long, e.g. lines 90-94 contain one sentence.

Author Response

Thanks for your suggestions. 

English has been revised and grammatical errors corrected.

Due to confidential aggreament with FAGOR, we cannot introduce the axis information. The values of the axis is from 0 to 1000 mm. Axis information has been adeed but not the values. We hope that this modification fits your expectations.

Reference to tables in result has been modified

Reviewer 5 Report

There is no further comments. Thank you

Minor editing of English language required

Author Response

Thanks for your comments.

English has been revised and grammatical errors corrected.